# OpenReview forum: "SCAM: A Real-World Typographic Robustness Evaluation for Multimodal Foundation Models"
_DMLR — Accepted by DMLR_

### Review · Reviewer_b5xH · 2025-11-03

**Recommendation:** 5
**Confidence:** 3

**Summary Of Contributions:**

This paper introduces a new datasets to test textual attacks on multi-model VLMs.
These datasets are the largest and allow to benchmark the impact of textual attacks
across all VLM models.

**Strengths:**

See above.

**Audience:**

Yes

**Claims And Evidence:**

All claims are experimentally verified.

**Datasets And Benchmarks:**

There are no ethical concerns here. The dataset was painstakingly gathered and the pipeline is well documented. A thanks to the authors for doing this.

**Extended Submissions:**

N/A

**Limitations:**

See above.
The analysis is precisely valid for one prompt and the given 1162 samples (maybe x2 given the synthetic and hand written dataset).

**Requested Changes:**

If possible, exploring different prompt templates, maybe also those not mentioning the attack word
might be helpful. That said, I do not think this should delay the publication of this paper as the community
benefits more from having access to this benchmark dataset soon and can always proceed with different
prompts on its own.

**Strengths And Weaknesses:**

Strengths:
1. Clearly written with clear goal.
2. Impactful results.
3. Opens up a new benchmark.

Weakness:
1. In a way, the datasets (SCAM and SynthSCAM) are still small so one can always argue about the
composition of subclasses chosen.
2. The evaluation is done wrt a fixed prompt displayed on page 8 which contains the attack word. A natural
question would be to extend this to different prompts that maybe do not mention the attack word.

---

### Review · Reviewer_wKkR · 2025-11-10

**Recommendation:** 3
**Confidence:** 1

**Summary Of Contributions:**

This paper introduces SCAM, a novel dataset designed to benchmark the robustness of Vision-Language Models (VLMs) against typographic attacks. To thoroughly assess attack effectiveness, the authors present three distinct evaluation setups: the Real-world SCAM dataset, a cleaned baseline (NoSCAM), and a dataset of digitally simulated attacks (SynthSCAM). The experiments demonstrate that both real-world (handwritten) and synthetic typographic attacks remain a significant threat to VLMs.

**Strengths:**

See Strengths And Weaknesses.

**Audience:**

Yes

**Claims And Evidence:**

Yes.

**Datasets And Benchmarks:**

The authors provided data collection details and the dataset is published on hugging face. However, the authors did not provide maintenance plan or ethical and responsible use of the data.

**Extended Submissions:**

The link in the paper suggested that this paper was accepted at CVPR 2025 EVAL-FoMo 2 workshop. However, I did not find the original version of the workshop paper (only an arxiv version exists), so I can not comment on whether it meets the eligibility criteria.

**Limitations:**

See Strengths And Weaknesses.

**Requested Changes:**

W1-W3

**Strengths And Weaknesses:**

S1. The idea of using synthetic attacks make dataset creation in the future much easier.
S2. The authors conduct comprehensive experiments to benchmark existing VLMs and and suggested a few interesting observations.
S3. The paper is well written and easy to follow.

W1. The contribution is not significant enough to me. What kind of research / new findings can SCAM enable while the previous datasets cannot? In the experiment section, the authors mention "Both evaluations are conducted on the SCAM datasets (SCAM, NoSCAM,
SynthSCAM) and additionally on PAINT (Ilharco et al., 2022) and RTA-100 (Azuma and Matsui, 2023)." However, the main result section only contains the result with SCAM. The result in appendix a.3 seems like the result of scam is quite similar to paint and RTA-100.
W2. It would be better to show how are the attacks chosen. Does the attack word matter?
W3. The dataset lacks maintenance plan and ethical and responsible use of the data.

---

### Review · Reviewer_2oix · 2025-11-24

**Recommendation:** 4
**Confidence:** 2

**Summary Of Contributions:**

The Authors have claimed to create the largest diverse dataset of real-world typographic attack images. It includes object categories, attack words, and aligned variants. They have benchmarked against VLMs and LVLMs, showing degradation under typographic attacks.

I am inclined to consider this as a good contribution.

**Strengths:**

same as above

**Audience:**

Yes

**Broader Impact Concerns:**

No concerns

**Claims And Evidence:**

yes, but please see weakness point 4

**Datasets And Benchmarks:**

Dataset is hosted on HuggingFace with sufficient details.

**Extended Submissions:**

NA

**Limitations:**

same as above

**Requested Changes:**

I would like to see more clarity, especially on the weak points 3 and 5

**Strengths And Weaknesses:**

Strengths:
1. The dataset is huge and diverse. (1162 - 660 obj categ, and 206 attack wrds - 1,147 unique object-word combinations)
2. Identical scenes can help isolate the effect of the attack text.
3. Benchmarking against open source models
4. high correlation with synthetic evaluation

weakness:
1. The testing of Large Vision-Language Models (LVLMs) only used a basic question format. This simple approach might exaggerate or change the true vulnerability of the models.
2. The diversity of attacks is limited and doesn't cover other visual threats like different fonts, placements, or materials.
3. I would have expected some reasoning/ evaluation behind why models are failing. Focus on attention mechanism/ Embedding space etc.
4. Comparison to previous work is purely quantitative, can you do some qualitative analysis as well.
5. How did you reach to conclusion that vision encoder is problem?